Integrated transcriptome and metabolome analyses revealed regulatory mechanisms of flavonoid biosynthesis in Radix Ardisia

Liu Chang 1
Pan Jie 1
Yin Zhi-Gang 2
Feng Tingting ftt0809@163.com 1
Zhao Jiehong 1
Dong Xiu 3
Zhou Ying yingzhou71@sina.com 1 2
1 School of Pharmacy, Guizhou University of Traditional Chinese Medicine , Guiyang , China
2 Guizhou Engineering Center for Innovative Traditional Chinese Medicine and Ethnic Medicine, Guizhou University , Guiyang , China
3 Guizhou Sanli Pharmaceutical Co., Ltd. , Guiyang , China
Fukushima Atsushi
Electronic publication date: 2022 Jun 29
Publication date: 2022
Volume: 10
Electronic Location ID: e13670
Received 2021 Sep 30; Accepted 2022 Jun 10
Copyright: ©2022 Liu et al.
Copyright year: 2022
Copyright holder: Liu et al.
License: This is an open access article distributed under the terms of the Creative Commons Attribution License, which permits unrestricted use, distribution, reproduction and adaptation in any medium and for any purpose provided that it is properly attributed. For attribution, the original author(s), title, publication source (PeerJ) and either DOI or URL of the article must be cited.
License URL: https://creativecommons.org/licenses/by/4.0/

Keywords: Ardisia crenata Sims, Ardisia crispa (Thunb.) A.DC., Transcriptomics, Metabolomics, Conjoint analysis, Flavonoid biosynthesis

Funding: National key research and development programs 2018YFC1708100 Science and Technology Plan Project of Guizhou QKHRCPT [2019]5407 Science and Technology Support Program of Guizhou Province No. 2019-2776 Guizhou Province “Hundred” Innovative Talents Project QKHRC(2015)4032 Guizhou Provincial Department of Science and Technology Academic Seedling Project QKHPTRC[2018]5766-9 Guizhou University of Traditional Chinese Medicine Scientific Research Innovation and Exploration Special Project 2018YFC170810201 2018YFC170810205 2018YFC170810101 Guizhou University of Traditional Chinese Medicine Doctor foundation [2019]04 This work was supported by National key research and development programs (2018YFC1708100), the Science and Technology Plan Project of Guizhou (QKHRCPT [2019]5407), the Science and Technology Support Program of Guizhou Province (No. 2019-2776), the Guizhou Province “Hundred” Innovative Talents Project (QKHRC(2015)4032), the Guizhou Provincial Department of Science and Technology Academic Seedling Project (QKHPTRC[2018]5766-9), the Guizhou University of Traditional Chinese Medicine Scientific Research Innovation and Exploration Special Project (2018YFC170810201; 2018YFC170810205; 2018YFC170810101), and the Guizhou University of Traditional Chinese Medicine Doctor foundation ([2019]04). The funders had no role in study design, data collection and analysis, decision to publish, or preparation of the manuscript.

==============================
Background

Radix Ardisia (Jab Bik Lik Jib) is a common Miao medicine and is widely distributed in the Guizhou region of southern China. The botanical origin of Radix Ardisia includes the dry root and rhizome of Ardisia Crenata Sims (ACS) or Ardisia Crispa (Thunb.) A.DC. (AC), which are closely related species morphologically. However, the secondary metabolites in their roots are different from one another, especially the flavonoids, and these differences have not been thoroughly explored at the molecular level. This project preliminarily identified regulatory molecular mechanisms in the biosynthetic pathways of the flavonoids between ACS and AC using a multi-omics association analysis.

Methods

In this study, we determined the total levels of saponin, flavonoid, and phenolic in Radix Ardisia from different origins. Integrated transcriptome and metabolome analyses were used to identify the differentially expressed genes (DEGs) and differentially expressed metabolites (DEM). We also performed conjoint analyses on DEGs and DEMs to ascertain the degree pathways, and explore the regulation of flavonoid biosynthesis.

Results

The total flavonoid and phenolic levels in ACS were significantly higher than in AC (P < 0.05). There were 17,685 DEGs between ACS vs. AC, 8,854 were upregulated and 8,831 were downregulated. Based on this, we continued to study the gene changes in the flavonoid biosynthesis pathway, and 100 DEGs involving flavonoid biosynthesis were differentially expressed in ACS and AC. We validated the accuracy of the RNA-seq data using qRT-PCR. Metabolomic analyses showed that 11 metabolites were involved in flavonoid biosynthesis including: Naringenin, Luteolin, Catechin, and Quercetin. A conjoint analysis of the genome-wide connection network revealed the differences in the types and levels of flavonoid compounds between ACS and AC. The correlation analysis showed that Naringenin, Luteolin, Catechin, and Quercetin were more likely to be key compounds in the flavonoid biosynthesis pathway also including 4CL, AOMT, CHS, CHI, DFR, F3’5’H, FLS, and LAR.

Conclusions

This study provides useful information for revealing the regulation of flavonoid biosynthesis and the regulatory relationship between metabolites and genes in the flavonoid biosynthesis pathway in Radix Ardisia from different origins.

Introduction

Radix Ardisia (hmong: Jab Bik Lik Jib, also called Ba Zhua Jin Long) is a common Miao herbal medicine in Guizhou, and is recorded in “Tujing Materia Medica” “Tianbao Materia Medica” and “Guizhou Herbal Medicine.” The meridian distribution of Miao medicine is determined by property and flavor; hot drugs with sweet, hot, fragrant and spicy taste are classified as cold meridian, while cold drugs with sour, bitter and wet taste are classified as hot meridian. Radix Ardisia is cold in nature and belongs to the heat meridian category. It clears heat, detoxifies, and is regarded as a laryngological medicine by the Miao. Radix Ardisia is widely used in many prescription formulas including Kaihou Jian Spray and Yangyin Kouxiang Mixture, and has a high medicinal value. The botanical origin of Radix Ardisia includes both the dry root and rhizome of Ardisia Crenata Sims, Ardisia Crispa (Thunb.) A.DC. and Ardisia Crenata Sims var. bicolor (Walker) C. Y. Wu et C. Chen, which are closely related species morphologically, but their chemical compositions and content are not exactly the same. The multiple origins leads to mixed varieties of medicinal materials, which affects both the quality and medicinal safety of Radix Ardisia.

Ardisia Crenata Sims (ACS, Chinese name “Zhu Sha Gen”), Ardisia Crispa (Thunb.) A.DC. (AC, known as “Bai Liang Jin”) and Ardisia Crenata Sims var. bicolor (Walker) C. Y. Wu et C. Chen (ACSV, known as “Hong Liang San” varieties of Ardisia Crenata Sims) belong to the genus Ardisia of the Myrsinaceae family (Song & Wu, 1988), which is a widely distributed shrub in the Guizhou region of southern China. ACS and AC were first recorded in the “Ben Cao Gang Mu” (Ming Dynasty in China), and the roots of the ACS and AC have been traditionally used to treat various diseases (Muhammad & Mustafa, 1994; Kobayashi & de Mejia, 2005; de Meija & Ramirez-Mares, 2011). Pharmacological studies have shown that AC exhibits anti-inflammatory, anti-arthritic, and anti-tumor activity, and can inhibit angiogenesis (Wen Jun, Pit Foong & Abd Hamid, 2019; Hamid, Fong & Ting, 2018; Sulaiman et al., 2012; Hamsin et al., 2013). Pharmacological studies have also shown that ACS has anti-tumor, anti-human immunodeficiency virus, anti-viral, and anti-oxidative effects (Zhang et al., 2010).

The compounds reported in Radix Ardisia include triterpene saponins, flavonoids, and coumarins (Kobayashi & de Mejia, 2005; Liu et al., 2007; Zheng et al., 2008; Liu et al., 2016; Li et al., 2021; Jansakul, Herbert & Lennart, 1987; Yoshida, Koma & Kikuchi, 1987; Kang et al., 2001; Ma et al., 2015), with the content of the main component (bergenin) different in ACS and AC (Wang et al., 2020). The 17 flavonoids and 10 coumarins were separated from different origins in Radix Ardisia. There were five common components, 22 different chemical components, and the contents of the common components were significantly different (Li et al., 2021); the secondary metabolites in their roots were different, especially the flavonoids. Differences in the types and contents of chemical components in medicinal materials may lead to differences in the efficacy of different botanical origins.

Flavonoids are naturally occurring compounds that display many pharmacological effects such as anti-tumor, antioxidant, and anti-inflammatory activities (Maleki, Crespo & Cabanillas, 2019; Maleki, Crespo & Cabanillas, 2019; Zhang et al., 2021). Flavonoids are derivatives of chalcones, including flavanones (naringenin), flavonols (quercetin), and isoflavones (formononetin; Aoki, Akashi & Ayab, 2000; Winkel-Shirley, 2001). Flavonoids can control mediators involved in inflammation by restraining regulatory enzymes or transcription factors (Maleki, Crespo & Cabanillas, 2019). Flavonoid biosynthesis begins with the phenylpropane metabolic pathway in plants, with the genes taking part in the flavonoid biosynthetic pathway classified into two categories (Weisshaar & Jenkins, 1998; Dixon et al., 2002; Nabavi et al., 2020). RNA sequencing (RNA-seq) has been widely used to study the regulatory mechanisms of flavonoid biosynthetics, such as Artemisia Annua L (Liu et al., 2017), Saussurea Lappa (Bains et al., 2019), Tetrastigma Hemsleyanum (Peng et al., 2019), and Pueraria Thomsonii Benth (He et al., 2019). The types and contents of flavonoid components are different in ACS and AC, but the regulatory mechanism of flavonoids is unknown and has been poorly explored at the molecular level.

Multi-omics association analysis has become a widely used biological method for genome analysis (Szymanski et al., 2014; Chen et al., 2012), so RNA-seq was performed to investigate the differentially expressed genes in the two origins of Radix Ardisia (ACS and AC). This RNA-seq analysis provided key insights into the regulatory mechanisms of flavonoid biosynthesis. We then used UHPLC-QTOF-MS techniques to scan the metabolites, and to investigate the regulatory relationship between genes and flavonoid biosynthesis, as well as screen the biomarkers of ACS and AC. The results have great significance in our understanding of the metabolic pathway of flavonoid biosynthesis and may help establish effective quality classification and evaluation methods.

Material and Methods

Determination of total saponin, flavonoid, and phenolic content

Ardisia Crenata Sims (ACS) and Ardisia Crispa (Thunb.) A.DC. (AC) plants were collected in Guiyang County, Guizhou Province, China (N.109.437569, E.19.19680). The samples were authenticated by Professor Sheng Hua Wei and stored at the Laboratory of Traditional Chinese Medicine and Ethnic Medicine, Guizhou University of Traditional Chinese Medicine, Guiyang, Guizhou, China. The specimen accession number was GUTCM0059.

Total saponin content (TSC) was measured according to the method described by Medina-Meza et al. (2016) with minor modifications. 1 mL of ACS or AC extract was added to a 10 mL test tube, and then 0.2 mL 5% Vanillin acetic acid solution and 0.8 mL HCLO4 were added after 6 min, vortexed vigorously, and heated at 60 °C for 10 min; after the mixture was cooled in ice water, five mL glacial acetic acid was added. The absorbance was measured at 545 nm. Glacial acetic acid was used as the blank. The TSC was obtained by comparison with the standard curve of oleanolic acid; the analysis was done in triplicate.

Total flavonoid content (TFC) was determined according to the method described by Xu et al. (2014) with some modifications. 1 mL Ardisia Crenata Sims (ACS) or Ardisia Crispa (Thunb.) DC (AC) extract was added to a 10 mL test tube, and then 2.4 mL 70% ethyl alcohol and 5% NaNO2 (0.4 mL) were added after 6 min. Then, 0.4 mL 10% Al (NO3)3 was added. After 6 min, 4 mL 4% NaOH and 70% ethyl alcohol were added at room temperature for 15 min. It was then measured against ethyl alcohol as the blank at 510 nm, a calibration curve of rutin was plotted to calculate the TFC, and the analysis was done in triplicate.

The analysis of total phenolic content (TPC) was carried out according to the Folin-Ciocalteau spectrophotometric method, with some modifications (Vieira de Morais et al., 2021). The 1 mL gallic acid (standard solution) or ACS and AC extract and 3 mL Folin–Ciocalteau solution (10% in water) were pipetted into a 25 mL test tube. After 5 min, 6 mL 10% sodium carbonate aqueous solution was added to each tube. A control was prepared by replacing the sample with distilled water. The absorbance was measured at 750 nm after 50 min. The TPC was calculated by linear regression using gallic acid as the standard; all samples were analyzed in triplicate.

RNA extraction, library preparation and sequencing

In this study, 12 root samples produced from Ardisia Crenata Sims (ACS) and Ardisia Crispa (Thunb.) DC (AC) (six biological replicates) were tested using RNA-seq. Total RNA was extracted with the RNAprep Pure Plant Kit (Tiangen, China) and detected on 1% agarose gel. The total RNA concentration was determined using an Agilent 2100 Bioanalyser, and the purity of the samples was measured using a NanoDrop™ ultraviolet spectrophotometer. The purity, concentration, and integrity of the total RNA samples were assessed prior to further analysis. After the isolation and fragmentation of the total RNA, mRNA was enriched using poly-T oligo-attached magnetic beads. The enriched mRNA was fragmented into short fragments using M-MuLV Reverse Transcriptase and reverse transcribed into cDNA using random hexamer primer. Second strand cDNA was synthesized using DNA Polymerase I and RNase H. Then, 3 µl USER Enzyme (NEB, USA) was used with size-selected, adaptor-ligated cDNA at 37 °C for 15 min followed by 5 min at 95 °C before PCR. Finally, PCR products were purified (AMPure XP system) and library quality was assessed on the Agilent Bioanalyzer 2100 system. After cluster generation, the library preparations were sequenced using an Illumina HiSeqTM 4000 platform by Biomarker Technologies (Beijing, China). In this work, twelve cDNA libraries produced from ACS and AC were sequenced using the Illumina HiSeq™ 4000 platform and 150bp paired-end reads were generated. Clean reads were obtained by removing lower quality reads as well as raw reads containing an adapter or ploy-N. Transcriptome assemblies for the twelve libraries were performed separately using Trinity (Garber et al., 2011) with min_kmer_cov set to two by default and all other parameters set to default.

DEGs related to secondary metabolism pathways

Gene expression levels were calculated using RSEM (Li & Dewey, 2011) for each sample. The differentially expressed genes (DEGs) were identified by comparing raw readcounts using the DESeq 2 (Love, Huber & Anders, 2014). Genes with FC ≥ 1.0 and p-value <0.05 were identified as DEGs. We used the OmicShare tools (https://www.omicshare.com/tools) and KOBAS 2.0 (Xie et al., 2011) software to test the statistical enrichment of DEGs in KEGG pathways, and the significance of KEGG terms was assessed using the Bonferroni corrected Fisher exact test (P < 0.05). We then selected the pathways associated with flavonoid biosynthesis for a more detailed analysis.

qRT-PCR verification

In order to verify the accuracy of the RNA-seq, a qRT-PCR analysis was used to assess the quality of the RNA-seq data. Total RNA was extracted from root tissue with TRIzol according to the manufacturer’s instructions (TIANGEN, China), and the isolated RNA was reverse-transcribed into cDNA with the RevertAid First Strand cDNA Synthesis Kit (Thermo Fisher, China). The expression of randomly selected genes was monitored by qRT-PCR using the SYBR Green qPCR Mastermix (TIANGEN, China) real-time PCR system by Bio-Rad CFX 96™ following the manufacturer’s instructions. Detailed information about the primer sequences for qRT-PCR was provided by the primer3 platform (http://frodo.wi.mit.edu/primer3/; Table S6). The qRT-PCR was used on three biological replicates with the expression levels of the genes determined using the 2−−ΔΔCT method. Expression levels were normalized against the GAPDH. Data are represented as mean values  ± standard deviation, and the GraphPad Prism 6 software was employed to draw the histogram.

Metabolite profiling using UHPLC-QTOF-MS

In this study, 12 samples from ACS and AC (six biological replicates, respectively) were scanned for metabolite determination by UHPLC-QTOF-MS, and the samples were consistent with RNA-seq.25 mg of root tissue sample after liquid nitrogen grinding was weighed and placed in a 1.5 ml EP tube, and 500 µL extract solution (acetonitrile: methanol: water = 2: 2: 1) containing isotopically-labelled internal standard mixture was added to each EP tube. After a 30s vortex, the samples were placed in the TissueLyser at 35 Hz for 4 min and sonicated for 5 min in an ice-water bath. The samples were then incubated for 1 h at −40 °C and centrifuged at 12,000 rpm for 15 min at 4 °C. The supernatant was transferred to a fresh EP tube and dried at room temperature. Then, the samples were reconstituted in 300 µL 50% acetonitrile by sonication for 10 min, then centrifuged at 13,000 rpm for 15 min at 4 °C, and 75 µL of supernatant was transferred to a fresh glass vial for LC-MS analysis.

Metabolomics data analysis

The UHPLC-MS/MS analyses were applied to the 1290 Infinity series UHPLC System (Agilent Technologies) coupled with a TripleTOF 6600 mass spectrometer (AB Sciex) from Biomarker Technologies (Beijing, China). The MS raw data were converted to the mzXML format by ProteoWizard, and processed by R package XCMS (version 3.2). This process included peak deconvolution, alignment and integration. Minfrac and cut off were set as 0.5 and 0.3, respectively. An in-house MS/MS database was applied for metabolites identification. The principal component analysis (PCA) and partial least squares discriminant analysis (PLS-DA) were performed using R. We employed a univariate analysis to calculate statistical significance; metabolites with VIP ≥ 1, fold change ≥ 2.0 or ≤ 0.50, and a p-value <0.05, were considered differentially expressed metabolites (DEM). Volcano plots were used to filter the metabolites of interest based on the log2 (FC) and log10 (p-value) of the metabolites. The KEGG database was used to annotate the differentially expressed metabolites with a p-value less than 0.05 set as the threshold. The enrichment factor represented the ratio between the proportion of differentially expressed metabolites in the pathway and all the metabolites in the pathway; the greater the value, the greater the degree of enrichment.

Combined transcriptome and metabolome analyses

To uncover the regulatory mechanism of flavonoid biosynthesis, a correlation analysis was performed using Pearson’s correlation coefficient to calculate the correlation coefficient (CC). The p-value of DEG and DEM, and the —CC—>0.80 and CCP <0.05 were used for the cluster analysis. The correlation matrix was imported into the Cytoscape software to visualize the DEG and DEM network. The canonical correlation analysis (CCA) is statistical technique for studying associations between two sets of variables (Hotelling, 1936), and applied to the integration of data originating from different “omics” technologies (Le Cao, Gonzlez & Djean, 2009).

Results

Total saponin, flavonoid, and phenolic content in Radix Ardisia from different origins

The total saponin content (TSC), total flavonoid content (TFC), and total phenol content (TPC) in Radix Ardisia from different origins were measured using the colorimetric method. The TSC in ACS was markedly higher than in ACSV (P < 0.05), however, there was no difference between the TSC in ACS and AC. The TFC and TPC in ACS were dramatically higher than in AC (P < 0.05) (Fig. 1). The differences in gene expression at varying origins in Radix Ardisia may lead to differences in the accumulation of secondary metabolites. We performed RNA-seq to examine the differences in gene expression between ACS and AC, and explore the regulation of flavonoid biosynthesis.

Figure 1 Total saponin, flavonoid, and phenolic content in Radix Ardisia from different origins (mg/g).

ACS, Ardisia Crenata Sims; ACSV, Ardisia Crenata Sims var. bicolor (Walker) C. Y. Wu et C. Chen; AC, Ardisia Crispa (Thunb.) A.DC.

RNA sequencing and assembly results

We compared transcriptional profiles, using RNA-Seq. The root samples from ACS and AC were each sequenced in six replicates. Twelve RNA-Seq libraries of root tissue generated approximately 89.73 GB of clean data. The high-quality reads totaled 52,249 Unigenes, with an average length of 1440 bp; N50 was 2336 bp, the alignment rate was more than 67.52%, and the Q30 base percentages were greater than or equal to 94.85% (Table 1). The results indicated that the assembly quality was reliable, and satisfactory for further analysis. The raw sequencing data were deposited in the NCBI Bioproject database under accession number PRJNA739135. The Unigenes data were deposited in the Transcriptome Shotgun Assembly(TSA) Database under accession number GJZC00000000, and SUBID was SUB11534240.

Table 1 Throughput and quality of RNA-seq of samples.

Sample-ID	Clean reads	Mapped reads	Mapped ratio	GC content	% ≥Q30	
ACS01	27,425,579	18,201,986	66.37%	45.67%	94.85%	
ACS02	26,562,212	17,749,901	66.82%	45.63%	94.95%	
ACS03	22,037,780	15,166,862	68.82%	45.65%	95.15%	
ACS04	22,346,460	15,078,022	67.47%	45.90%	94.66%	
ACS05	19,247,338	13,063,564	67.87%	45.23%	95.27%	
ACS06	22,099,726	15,023,313	67.98%	45.93%	95.02%	
AC01	26,117,088	17,508,303	67.04%	45.80%	94.92%	
AC02	22,726,418	15,200,905	66.89%	45.77%	94.77%	
AC03	30,932,371	20,521,520	66.34%	46.24%	95.30%	
AC04	26,259,169	17,721,761	67.49%	45.93%	95.10%	
AC05	26,490,511	18,084,432	68.27%	45.79%	95.33%	
AC06	28,251,394	19,462,934	68.89%	45.87%	95.29%  	

Identification of differentially expressed genes

A total of 52,249 Unigenes were identified, which were expressed in at least one sample (Table S1). DESeq 2 was used to analyze the differentially expressed genes (DEGs) in Ardisia Crispa (Thunb.) DC. from origins ACS and AC. There were 17,685 DEGs between ACS vs. AC: 8,854 were upregulated and 8,831 were downregulated (Table S2; Fig. 2). To confirm the reliability of the gene expression, 20 genes were randomly selected for quantitative real-time polymerase chain reaction (qRT-PCR) detection. The qRT-PCR showed that the tendency of gene expression was similar to the RNA-Seq results (Figs. 3A and 3B). The results showed that the RNA-Seq results were reliable in this study.

Figure 2 The differentially expressed genes between ACS and AC (A) Volcano plot; (B) MA plot.

Red dots represent upregulated DEGs and green dots represent downregulated DEGs.

Figure 3 The qRT-PCR analysis of the genes.

(A) The RNA-seq results revealed differentially expressed genes. (B) Differentially expressed genes were confirmed by qRT-PCR. Data are represented as mean values ± SD. n= 3.

DEG functional enrichment analysis

A KEGG pathway analysis showed that 139 pathways were represented in the transcriptome dataset (Table S3). Interestingly, these DEGs in ACS and AC were mainly enriched in the metabolism, genetic information processing, environment information processing, cellular process, and organismal systems, involved in Inositol phosphate metabolism, Other glycan degradation, Homologous recombination, Mismatch repair, Phosphatidylinositol signaling system, Glycosylphosphatidylinositol (GPI)-anchor biosynthesis, Autophagy - other, Ubiquitin mediated proteolysis, Basal transcription factors, Galactose metabolism, mRNA surveillance pathway, Glycosaminoglycan degradation, Butanoate metabolism, Fructose and mannose metabolism, Phenylalanine, tyrosine and tryptophan biosynthesis, Circadian rhythm - plant, Starch and sucrose metabolism, Nicotinate and nicotinamide metabolism, and Taurine and hypotaurine metabolism (P < 0.05) (Fig. 4; Table 2).

Figure 4 The main terms of the KEGG analysis.

Table 2 The main terms of KEGG analysis (P < 0.05).

Pathway ID	Pathway	Out	All	P value	
ko00562	Inositol phosphate metabolism	70	94	8.59E−05	
ko00511	Other glycan degradation	25	31	0.002874676	
ko03440	Homologous recombination	59	84	0.003384125	
ko03430	Mismatch repair	45	62	0.003718499	
ko04070	Phosphatidylinositol signaling system	65	95	0.00579574	
ko00563	Glycosylphosphatidylinositol (GPI)-anchor biosynthesis	27	35	0.006143562	
ko04136	Autophagy - other	44	62	0.008039568	
ko04120	Ubiquitin mediated proteolysis	138	219	0.01107059	
ko03022	Basal transcription factors	48	70	0.01559127	
ko00052	Galactose metabolism	59	88	0.01576872	
ko03015	mRNA surveillance pathway	128	204	0.01695472	
ko00531	Glycosaminoglycan degradation	14	17	0.01930863	
ko00650	Butanoate metabolism	25	34	0.02230163	
ko00051	Fructose and mannose metabolism	63	96	0.0245332	
ko00400	Phenylalanine, tyrosine and tryptophan biosynthesis	46	68	0.02496569	
ko04712	Circadian rhythm - plant	46	68	0.02496569	
ko00500	Starch and sucrose metabolism	119	191	0.02746657	
ko00760	Nicotinate and nicotinamide metabolism	26	36	0.02796564	
ko00430	Taurine and hypotaurine metabolism	16	21	0.04073509	

DEGs related to flavonoid biosynthesis

DEGs involved in the flavonoid biosynthetic pathway included those integral to phenylpropanoid biosynthesis (ko00940, 52 genes), phenylalanine metabolism (ko00360, 26 genes), flavonol and flavone biosynthesis (ko00944, three gene), flavonoid biosynthesis (ko00941, 32 genes) and anthocyanin biosynthesis (ko00942, one gene). Table S4 showed that 100 DEGs (49 upregulated and 61 downregulated) were differentially expressed in ACS and AC. Furthermore, eight chalcone synthase (CHS, Unigene_171144, Unigene_186235, Unigene_108709, Unigene_182412, Unigene_168887, Unigene_030909, Unigene_001289, Unigene_029241), one chalcone isomerase (CHI, Unigene_026449), six dihydroflavonol 4-reductase (DFR, Unigene_030972, Unigene_181720, Unigene_106630, Unigene_173665, Unigene_092658, Unigene_180983), two flavonoid 3′5′-hydroxylase (F3′5′H, Unigene_094347, Unigene_094346), one flavanone 3-hydroxylase (F3H, Unigene_026466), two flavonol synthase (FLS, Unigene_013848, Unigene_015774), one leucoanthocyanidin reductase (LAR, Unigene_174994), one phenylalanine ammonia-lyase (PAL, Unigene_177796), four 4-coumarate: CoA ligase (4CL, 008420, Unigene_171229, Unigene_025326, Unigene_007828), and two cinnamate 4-hydroxylase (C4H, Unigene_181127, Unigene_013550, not significant) were identified in the second developmental stage (Fig. 5).

Figure 5 The differentially expressed genes of flavonoid biosynthesis in ACS and AC.

Metabolic differences in ACS and AC

RNA-Seq results demonstrated significant differences in metabolism between ACS and AC. Therefore, we investigated the changes in the metabolic compositions of ACS and AC. In this study, we used 12 samples to survey the differences in the metabolic constituents of ACS and AC with six biological replicates. First, the principal component (PC1) in ESI+ mode (34% of the total variables) and PC1 in ESI − mode (47.20%) were clearly separated between the ACS and AC groups (Figs. S1A and S1B). The VIP values of the first two principal components of the multivariate orthogonal partial least square discriminant (OPLS-DA) (Figs. S1C and S1D), and VIP ≥ 1, fold change ≥2.0 or ≤ 0. 50, and p-value <0.05 to screen for differentially expressed metabolites (DEM). A total of 943 and 1,250 DEMs were identified between the ACS and AC groups in the ESI+ (ESI −) mode, respectively (Table 3; Fig. 6).

Metabolic pathway analysis

A DEM analysis and KEGG pathway annotation of metabolites revealed that 124 DEMs were enriched in a variety of functional pathways (Table S5), including: flavonoid biosynthesis, phenylalanine metabolism, zeatin biosynthesis, indole alkaloid biosynthesis, isoflavonoid biosynthesis, ateroid biosynthesis, flavone and flavonol biosynthesis, fatty acid biosynthesis, phenylpropanoid biosynthesis, phenylalanine metabolism, tyrosine metabolism, pyrimidine metabolism, glutathione metabolism, and glucosinolate biosynthesis (Figs. 7A and 7B). In this study, 11 metabolites were authenticated including L-Phenylalanine, p-Hydroxycinnamaldehyde, L-Tyrosine, Ferulic acid, Sinapic acid, Sinapyl alcohol, 4-Hydroxycinnamic acid, Naringenin, Luteolin, Catechin, and Quercetin, all of which participated in flavonoid biosynthesis and the upstream phenylpropane metabolic pathway (Fig. 7C). Compared with ACS, the expression levels of six metabolites (L-Phenylalanine, p-Hydroxycinnamaldehyde, L-Tyrosine, Ferulic acid, Sinapic acid, Sinapyl alcohol) were significantly increased in AC, while the expression levels of five metabolites (4-Hydroxycinnamic acid, Naringenin, Luteolin, Catechin, Quercetin) were significantly decreased in AC.

Table 3 The results of differential ions and identification at ACS and AC.

Mode	Diff ion number	Up	Down	MS2	
ESI+	943	438	505	140	
ESI−	1250	1026	224	157	

Figure 6 (A) Differentially expressed metabolites (ESI+); (B) differentially expressed metabolites (ESI-).

Figure 7 The KEGG pathways involving differentially expressed metabolites between ACS and AC.

(A) ESI+; (B) ESI-; (C) Differentially expressed metabolites in flavonoid biosynthesis.

Correlation analysis between RNA-seq and metabolites uncovers the regulatory pathway of flavonoid biosynthesis

To explore the correlation between gene expression and metabolites, we performed correlation analyses of the metabolites related to flavonoid biosynthesis and the transcripts. The profiles of the metabolites and gene expression in ACS and AC were compared using a correlation coefficient and a canonical correlation analysis (CCA). These correlations and the CCA analysis results are shown in Figs. S2 and S3. The results indicated that metabolites such as Succinate, Salicylic acid, 2-Hydroxyphenylacetic acid, p-Hydroxycinnamaldehyde, L-Tyrosine, Phenylpyruvate, N-Acetyl-L-phenylalanine, Phenylacetic acid, Sinapic acid, Sinapyl alcohol, 4-Hydroxycinnamic acid, Alpha-N-Phenylacetyl-L-glutamine, Naringenin, 2-Phenylacetamide, Luteolin, Catechin, and Quercetin were more likely to be regulated in the flavonoid biosynthesis pathway. The correlation analysis between flavonoid-related genes and metabolites showed that 48 Key Unigenes (4CL, AOMT, CHS, CHI, DFR, LAR…) were significantly correlated with metabolites (Fig. 8; P < 0.05, —r— >0.8; Table 4). These results could provide insight into the relationship between genetically regulated metabolites and the metabolic impact on gene expression (Fig. 9).

Figure 8 Heat map of flavonoid-related genes and metabolites in ACS and AC.

449: Succinate; 1321: p-Hydroxycinnamaldehyde; 740: Salicylic acid (-); 2506: Sinapic acid; 1549: L-Tyrosine; 2138: N-Acetyl-L-phenylalanine; 1819: Ferulic acid; 3093: Alpha-N-Phen.

Discussion

In this study, a high-quality transcriptome database of Ardisia Crenata Sims (ACS) and Ardisia Crispa (Thunb.) A.DC. (AC) was generated based on RNA-seq technology to demonstrate the gene expression of Radix Ardisia from different origins. The reliability of the transcriptome results was verified using qRT-PCR. The metabolites of ACS and AC were generated using the UHPLC-QTOF-MS approach. The correlation analysis between the metabolites and genes was then performed to explore the regulation of flavonoid biosynthesis in ACS and AC.

The botanical origin of Radix Ardisia includes both ACS and AC, which are closely related species in appearance and structure, but their chemical composition and content are not completely the same. Because of the multiple origins of Radix Ardisia, there are many varieties of medicinal materials created from the plant, which can seriously affect the quality and medication safety of Radix Ardisia use. In the previous study, we used DNA barcoding to distinguish the original species from Radix Ardisia; the results showed that ITS sequencing can distinguish AC and ACS (Pan et al., 2020). The total flavone content (TFC) and total phenol content (TPC) in ACS was dramatically higher than in AC (P < 0.05). Previously, we employed the UPLC- QE- HF-MS/MS to identify and analyze the flavonoids in Radix Ardisia from different sources and were able to identify a total of 17 flavonoids, including nine flavonols, three flavane-3-alcohols and three other types of flavonoids (Li et al., 2021). The results also suggested that the difference in gene expression may lead to the difference in the synthesis and accumulation of flavonoid metabolites. Transcriptome and metabolome analyses have been widely used to study the regulatory mechanisms in the secondary metabolic biosynthetic pathways of medicinal plants, including Chamomile (Tai et al., 2020), Ginseng (Fan et al., 2019), Primula Oreodoxa (Zhao et al., 2019), Anoectochilus Roxburghii (Zhang et al., 2020), Rheum (Liu et al., 2020), Sophora Flavescens (Wei et al., 2021), Ziziphora Bungeana (He et al., 2020). Therefore, we performed comparative transcriptome and metabolome analyses to examine the differences in gene expression in ACS and AC, and explore the regulation of flavonoid biosynthesis.

Table 4 Identified 48 Key Unigenes significantly correlated with metabolite.

ID	Nr_annotation	ACS_FPKMavg	AC_FPKMavg	P value	log2FC	Regulated	
Unigene_001055	hypothetical protein	24.81 ± 14.67	2.58 ± 1.16	3.659E−14	−3.263895	down	
Unigene_005488	hypothetical protein	90.78 ± 30.64	0.04 ± 0.09	4.608E−50	−10.98121	down	
Unigene_007828	4-coumarate–CoA ligase-like 6	0.21 ± 0.16	2.23 ± 0.94	9.182E−09	3.3652447	up	
Unigene_009008	Peroxidase	0.14 ± 0.10	8.47 ± 4.49	1.461E−26	5.8751415	up	
Unigene_011439	Amidase	17.02 ± 5.89	0.73 ± 0.26	4.539E−36	−4.532826	down	
Unigene_011475	aldehyde dehydrogenase family 2 member C4-like isoform X1	139.74 ± 84.74	28.65 ± 21.36	3.773E−05	−2.327472	down	
Unigene_011573	Alcohol dehydrogenase, class V	22.87 ± 9.78	119.62 ± 32.97	3.827E−13	2.3600895	up	
Unigene_012260	Vrosine aminotransferase	9.51 ± 7.79	55.12 ± 32.27	2.995E−07	2.601553	up	
Unigene_012598	Alcohol dehydrogenase, class V	76.13 ± 42.70	33.44 ± 6.82	0.0016177	−1.199002	down	
Unigene_013498	Macrophage migration inhibitory factor homolog	11.99 ± 3.81	23.17 ± 9.81	0.0025813	0.9882798	up	
Unigene_013848	Flavonol synthase	17.66 ± 5.84	1.66 ± 0.94	1.138E−14	−3.435072	down	
Unigene_015370	LOC110664579	28.22 ± 21.42	1.39 ± 2.41	1.077E−07	−4.300268	down	
Unigene_019327	Histidinol-phosphate aminotransferase, chloroplastic-like	15.38 ± 5.38	0.26 ± 0.18	3.328E−39	−5.920615	down	
Unigene_021888	Peroxidase 11-like	0.19 ± 0.15	1.87 ± 2.34	0.0072191	3.0412866	up	
Unigene_021955	Cinnamoyl CoA reductase	6.73 ± 2.954	0.71 ± 0.41	2.693E−13	−3.270665	down	
Unigene_022250	beta-glucosidase-like protein	1.67 ± 0.74	30.56 ± 30.86	1.251E−07	4.1037958	up	
Unigene_026387	Coniferaldehyde 5-hydroxylase	37.36 ± 16.95	14.23 ± 7.38	0.0008897	−1.435376	down	
Unigene_026449	Chalcone isomerase	16.79 ± 10.23	2.21 ± 4.63	0.0044616	−2.800988	down	
Unigene_029550	beta-glucosidase 2 GH1 family	5.68 ± 1.50	0.10 ± 0.09	4.715E−30	−5.89852	down	
Unigene_030550	Cinnamyl alcohol dehydrogenase 1	27.36 ± 7.56	5.18 ± 2.86	1.306E−09	−2.390581	down	
Unigene_030972	Dihydroflavonol 4-reductase	1.99 ± 1.76	0.25 ± 0.25	0.000897	−3.035191	down	
Unigene_074510	Hypothetical protein	0.60 ± 0.87	3.50 ± 2.28	0.0025087	2.5791785	up	
Unigene_084303	Sinapyl alcohol dehydrogenase-like protein	387.62 ± 220.38	37.53 ± 26.40	1.791E−10	−3.287818	down	
Unigene_086160	Peroxidase 12	128.65 ± 45.93	5.99 ± 4.36	5.264E−16	−4.339712	down	
Unigene_092562	LOC104608192	38.20 ± 9.85	0.33 ± 0.38	1.484E−28	−6.731642	down	
Unigene_093188	beta-primeverosidase	2.00 ± 1.29	36.34 ± 43.35	3.167E−06	4.0486767	up	
Unigene_096163	Caffeic acid O-methyltransferase	101.88 ± 60.15	7.77 ± 5.43	4.853E−13	−3.661642	down	
Unigene_100382	Hypothetical protein	101.50 ± 63.46	18.54 ± 6.41	5.602E−07	−2.428365	down	
Unigene_105677	Macrophage migration inhibitory factor homolog	28.27 ± 6.86	3.43 ± 2.65	3.041E−10	−3.016909	down	
Unigene_106630	Dihydroflavonol reductase	1.99 ± 0.44	0.65 ± 0.389	0.0007112	−1.648286	down	
Unigene_106776	Peroxidase P7	9.49 ± 3.52	2.06 ± 3.27	0.0055049	−2.162008	down	
Unigene_107050	Aldehyde dehydrogenase family 2 member C4	52.89 ± 17.38	3.84 ± 1.67	1.064E−27	−3.766467	down	
Unigene_108709	Chalcone synthase	847.99 ± 524.29	273.87 ± 71.83	0.0004339	−1.688146	down	
Unigene_108847	Phenylalanine ammonia-lyase	138.51 ± 116.99	10.66 ± 5.27	5.33E−09	−3.679988	down	
Unigene_110196	AAT1/GOT2	1.16 ± 0.14	3.93 ± 1.04	5.021E−11	1.7506487	up	
Unigene_161437	–	2.67 ± 1.58	10.25 ± 7.09	0.0032382	1.8998033	up	
Unigene_171324	LOC104608192	9.07 ± 2.11	41.82 ± 12.63	1.605E−15	2.2059793	up	
Unigene_174437	Hypothetical protein	37.89 ± 9.28	9.85 ± 4.12	1.71E−08	−1.952215	down	
Unigene_175041	Transferase family protein	0.77 ± 0.59	26.23 ± 14.05	8.957E−23	5.0530583	up	
Unigene_175784	beta-glucosidase 9 GH1 family	0.00	2.27 ± 1.12	4.375E−08	8.9824046	up	
Unigene_175984	Cinnamoyl-CoA reductase	14.20 ± 6.01	45.15 ± 23.74	0.0002084	1.6085539	up	
Unigene_176701	Chalcone synthase 3-like isoform X1	153.43 ± 44.03	51.39 ± 52.71	0.0042192	−1.500815	down	
Unigene_177302	Alcohol dehydrogenase, class V	47.15 ± 26.80	107.56 ± 15.46	0.0003708	1.2380376	up	
Unigene_177521	Caffeic acid O-methyltransferase	5.86 ± 4.30	60.68 ± 20.83	1.073E−11	3.3448118	up	
Unigene_179620	beta-glucosidase 4 GH1 family	5.09 ± 2.24	16.38 ± 4.57	2.302E−08	1.7389418	up	
Unigene_180053	beta-glucosidase 1 GH3 family	5.42 ± 3.73	110.98 ± 73.95	3.062E−15	4.2259828	up	
Unigene_181720	Dihydroflavonol 4-reductase	102.23 ± 40.06	34.65 ± 26.79	0.0086711	−1.510286	down	
Unigene_186415	Caffeic acid 3-O-methyltransferase-like	0.02 ± 0.03	2.56 ± 0.98	1.33E−11	6.6685458	up	

Figure 9 Analysis of the genome-wide connection network between genes and metabolites related to flavonoids in ACS and AC.

The purple triangle represents metabolites and the blue circle represents genes.

The DEGs involved in flavonoid biosynthesis included those integral to phenylpropanoid biosynthesis, phenylalanine metabolism, flavonoid biosynthesis, flavonol and flavone biosynthesis and anthocyanin biosynthesis. There were 128 DEGs in ACS and AC identified in the second developmental stage, including CHS, CHI, DFR, F3′5′H, FLS, LAR, and PAL. Flavonoids are a large category of secondary metabolites ubiquitous in medicinal plants. Flavonoids biosynthesize through the phenylpropanoid pathway from substrate phenylalanine initially which is then converted to cinnamic acid through PAL catalysis (Ferrer et al., 2008; Li et al., 2020). We found that the expression of PAL genes was higher in ACS vs. AC. CHS is the key enzyme in the phenylpropane synthesis of flavonoids, and the expression of CHS in ACS was higher than AC. CHS in flavonoid biosynthesis can catalyze one molecule of p-cinnamoyl-CoA and three molecules of malonyl-CoA to produce naringenin chalcone. CHI can convert naringenin chalcone into naringenin, which can be converted by F3H into dihydrokaempferol (Qiang et al., 2020; He et al., 2018; Fig. 5).

In this study, we identified 11 metabolites involved in flavonoid biosynthesis, including the upstream phenylpropane metabolic pathway of flavonoid biosynthesis, containing L-Phenylalanine, p-Hydroxycinnamaldehyde, L-Tyrosine, Ferulic acid, Sinapic acid, Sinapyl alcohol, 4-Hydroxycinnamic acid, Naringenin, Luteolin, Catechin, and Quercetin. Compared with AC, the content of Naringenin, Luteolin, Catechin, and Quercetin were significantly increased in ACS. The regulatory network and CCA analysis showed that Naringenin, Luteolin, Catechin, and Quercetin were more likely to be key compounds in the flavonoid biosynthesis pathway, also including 4CL, AOMT, CHS, CHI, DFR, F3′5′H, FLS, and LAR. Naringenin, Luteolin, Catechin, and Quercetin, as flavanones from the flavonoids family, have shown anti-inflammatory, and antioxidant activities (Naraki, Rezaee & Karimi, 2021; Aziz, Kim & Cho, 2018; Nakano et al., 2019; Hou et al., 2019). The contents of the flavonoid components are different in ACS and AC, which may lead to differences in the efficacy of Radix Ardisia from different botanical origins. Further studies are needed to explore this possibility.

Conclusion

In summary, we found the total flavone content (TFC) and total phenol content (TPC) differed in the roots of ACS compared to AC. We also identified flavonoid biosynthesis-related genes with the majority more highly expressed in ACS than in AC. Furthermore, we explored the regulatory relationship between genes and flavonoid biosynthesis and metabolism. Correlation analyses can be very helpful for pinpointing candidate regulatory genes linked to compositional changes in ACS and AC. The data generated in the study will be an invaluable resource for further studies involving functional genomics, molecular biology, and plant breeding in ACS and AC.

Supplemental Information

Supplemental Information 1 PCA of positive ions; B: PCA of negative ions; C: PLS-DA score plots of positive ions; C: PLS-DA score plots of negative ions

Click here for additional data file.

Supplemental Information 2 Analysis of the canonical correlation analysis between regulatory genes and metabolites related to flavonoid biosynthesis in ACS and AC

Click here for additional data file.

Supplemental Information 3 Analysis of the connection network between regulatory genes and metabolites related to flavonoid biosynthesis in ACS and AC

Click here for additional data file.

Supplemental Information 4 Supplementary Information

Click here for additional data file.

Abbreviations

ACS Ardisia Crenata Sims

AC Ardisia Crispa (Thunb.) A.DC.

CHS chalcone synthase

CHI chalcone isomerase

DFR dihydrokaempferol 4-reductase

LDOX leucocyanidin oxygenase

BAN anthocyanidin oxidoreductase

PAL phenylalanine ammonia-lyase

C4H cinnamate 4-hydroxylase

4CL 4-coumarate CoA ligase

TSC total saponin content

TFC total flavonoid content

TPC total phenol content

DEGs Differentially Expressed Genes

DEM differentially expressed metabolites

KEGG Kyoto encyclopedia of genes and genomes

GO Gene Ontology

LC-MS liquid chromatograph-mass spectrometer

OPLS-DA orthogonal partial least squares discriminant

PCA principal component analysis

qRT-PCR quantitative real-time polymerase chain reaction

RNA-seq RNA-sequencing

VIP variable importance in project

CC correlation coefficient

CCA canonical correlation analysis

Additional Information and Declarations

Competing Interests

Author Contributions

Field Study Permissions

Data Availability

Xiu Dong is employed by Guizhou Sanli Pharmaceutical Co., Ltd.

Chang Liu conceived and designed the experiments, performed the experiments, analyzed the data, prepared figures and/or tables, authored or reviewed drafts of the article, and approved the final draft.

Jie Pan performed the experiments, analyzed the data, prepared figures and/or tables, and approved the final draft.

Zhi-Gang Yin performed the experiments, prepared figures and/or tables, and approved the final draft.

Tingting Feng conceived and designed the experiments, performed the experiments, authored or reviewed drafts of the article, and approved the final draft.

Jiehong Zhao analyzed the data, prepared figures and/or tables, and approved the final draft.

Xiu Dong analyzed the data, prepared figures and/or tables, and approved the final draft.

Ying Zhou conceived and designed the experiments, authored or reviewed drafts of the article, and approved the final draft.

The following information was supplied relating to field study approvals (i.e., approving body and any reference numbers):

Field experiments were approved by the Guizhou University of Traditional Chinese Medicine (project number: GUTCM0059).

The following information was supplied regarding data availability:

The raw sequencing data are available at NCBI: PRJNA739135.

The unigene data are available at the Transcriptome Shotgun Assembly (TSA) Database: Ardisia crenata, GJZC00000000.

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
