# Peer review of "Integrated transcriptome and metabolome analyses revealed regulatory mechanisms of flavonoid biosynthesis in Radix Ardisia"

_PeerJ, doi:10.7717/peerj.13670_

## Round 0.1 · original submission · Major Revisions

Dear Dr. Chang Liu and co-authors,

As you will see all the reviewers found the manuscript as potentially interesting but felt that it was little bit preliminary for publication in PeerJ. There was a significant gap between its current form of your manuscript and the publishable standard.

Reviewer 1 was stronger in their criticism and indicated a lack of detailed methods. Reviewer 2 also pointed out that the manuscript organization needs to be reworked considerably for clarity and to include missing details that make interpretation of the results difficult. Adding new experiments and/or data analysis will make your work stronger. I would be happy to handle a re-submission of a manuscript addressing all the issues.

Best regards

Reviewer 1 ·

Basic reporting

Some figures have low definition (e.g. Fig. 6). There is also a lot of room for improvement in writing.

Experimental design

I think it is probably an important contribution to the traditional medical field, but the authors could do a better job to make the results attractive to the general audience. In addition, the methods section can be elaborated to include many details.

Validity of the findings

no comment

Annotated reviews are not available for download in order to protect the identity of reviewers who chose to remain anonymous.

·

Basic reporting

Throughout the manuscript there are significant issues with English grammar and syntax to the point where it is sometimes difficult to understand the meaning of phrases. Aside from relatively minor problems such as missing direct and indirect articles (“a”, “the”) there are more serious problems with phrasing that make reading difficult. Below I have offered a few examples with suggestions for how to correct language issues. I would suggest, if possible, passing this manuscript along to a colleague with fluent English language mastery who can help to revise the writing throughout the paper. I believe this would help considerably with clarity and readability.

Lines 58-59: “Pharmacological studies showed that the AC possess anti-inflammatory, anti-arthritic, anti-tumor,
and inhibiting angiogenesis (Wen et al., 2019; Hamid et al., 2017; Sulaiman et al., 2012; Hamsin et al., 2013).”

Example of language revision: Pharmacological studies showed that AC exhibits anti-inflammatory, anti-arthritic, and anti-tumor activitity, and can inhibit angiogenesis.

Lines 71-73: “Flavonoids are naturally occurring compounds that possess many pharmacological effects, containing anti-tumor, anti-oxidant, anti-inflammatory and so on (Maleki et al., 2005; Maleki et al., 2019; Zhang et al., 2021).”

Example of language revision: Flavonoids are naturally occurring compounds that display many pharmacological effects such as anti-tumor, antioxidant, and anti-inflammatory activities.

Lines 109-110: “The analysis of total phenolic content (TPC) was used to Folin–Ciocalteau spectrophotometric method, with some modifications.” This statement is unclear to the point that I can’t tell exactly what it means.

Example of language revision: “The analysis of total phenolic content (TPC) was carried out according to the Folin–Ciocalteau spectrophotometric method, with some modifications.” I think this is what is meant here, but I am not 100% sure.

A related issue is that there a large number of typos such as missing spaces, extra articles (“the the...”), etc throughout the manuscript. I recommend a detailed review of the entire document to find and fix these issues.


Lines 41-43 of the introduction. Terms such as “heat meridian”, “stasis pain”, and “dispelling wind” are not familiar to me and I suspect they will not be familiar to many readers. I think these terms, and their origins, should be explained or that the the terms should be replaced with more commonplace medical terms. A further issue here is that there are not citations for the assertions made in this opening statement. I think it would be best to include some references that support these statements.

Lines 79-80 of the Introduction. Species names here and throughout the manuscript should be italicized to be consistent with the convention.

Lines 185-186 of the Results. It is stated that “The results indicated that the assembly quality was better in this study.” The assembly quality is better in comparison to what? Is this compared to another dataset? If so, that study should be cited here. Also, here the NCBI BioProject number for the raw reads is provided, which is great, but will the transcriptome assemblies also be made available somewhere for the sake of reproducibility?

Lines 196-205 of the Results (sub-header “DEG Function Enrichment Analysis”) is essentially just a single long list of ontology terms that were enriched in the KEGG pathways. This information would probably be better condensed into a table. Instead, this sub-section could be used to contextualize this information rather than simply listing categories.

Lines 285-289 of the Discussion. There is a fairly long discussion here of the steps in the upstream part of the flavonoid pathway. I think it would be very helpful for most readers to have a figure showing a simple diagram of the the pathway so that the logic is easier to visualize.

Lines 294-296 of the Discussion. It is stated that “The regulatory network and CCA analysis shown that Naringenin, Luteolin, Catechin, Quercetin was more likely to be key compound in the flavonoid biosynthesis pathway, involving in 4CL, AOMT, CHS, CHI, DFR, F3’5’H, FLS, LAR, and so on.” It is not clear to me what exactly is being argued here. How does the correlation analysis show that these flavonoids are more likely to be “key compounds” and what is the link to the list of genes that is provided? Is it saying that these flavonoids form a module (defined by correlation) with these particular pathway genes, which was identified by the CCA analysis? This needs to be rephrased and/or expanded to make the logic clearer.

Throughout the Methods section TFC is used to indicate “Total Flavonoid Content”, which makes sense to me, because a variety of flavonoids (including flavonols and flavones) are listed and discussed. However, through the Disscussion (and in the Abbreviations section) TFC is used to indicate “Total Flavone Content”. Based on the compounds that are discussed, I believe the authors still mean Flavonoid, but this should be checked and confirmed throughout the manuscript.

The image quality for some of the figures, such as Fig. 2, Fig. 4, and Fig. 6, is quite poor. This makes it difficult to see certain information. Also, I would recommend putting lines around the bars in the bar plots in Fig. 2, because the neighboring bars bleed together and are difficult to see clearly.

Figure captions, particularly for figures such as Fig. 3, Fig. 4, and Fig. 5, need to contain more detailed information about what is shown. These are complex figures and there is not enough information in the captions for readers to parse them efficiently. I also recommend declarative titles for the captions that state the takeaway messages.

In Fig. 7, the unigenes (blue ellipses) and the metabolites (red triangles? This information needs to be included in the captions!) are only labeled with numbers and unique IDs that are not meaningful to the reader. How should these results (e.g. the apparent clusters/modules in the network) be interpreted? It would also be very helpful to highlight the genes and metabolites that are elaborated on in the subsequent analysis and in the text (i.e. those that are associated with flavonoid biosynthesis). And again, the image quality of Fig. 7 is very poor and needs to be improved before incorporation into any final version of this manuscript.

I have a similar critique to Fig. 7 for Fig. 8. What is the identity of the genes and metabolites shown in this heatmap figure (i.e. which one is a CHS gene or the quercetin compound). Not having this information clearly communicated in the figure makes it extremely difficult to parse and to link what is being shown with the statements that are made in the text.

Experimental design

Lines 107-108 of the Methods. Why was methanol used as blank in this spec measurement? As far as I can tell from the description, there is no methanol in the solution used for the procedure? This should be explained. In this same procedure it is stated that “a calibration curve was plotted to calculated the TFC”. What was used to generate this calibration curve? A single flavonoid? A mixture of flavonoids? Different flavonoid compounds have very different absorption spectra, so it is necessary to know what was used for calibration in order to interpret the absolute values calculated from the calibration curve.

Lines 110-112 of the Methods. It is stated that “The 1 mL of gallic acid (standard solution) or ACS and AC extract and 3 mL of the Folin–Ciocalteau solution (10% in water) were pipetted into the wells of a microplate. After 5 min, 6 mL of a 10 % sodium carbonate aqueous solution were added to each well.” This does not seem to be possible. How can 6mL of solution fit into each well of a micro-plate? I am guessing this is supposed to be microliters?

Lines 117-118 of the Methods. It is stated that “In this study, 12 samples produced from Ardisia crenata Sims (ACS) and Ardisia Crispa (Thunb.) DC (AC) (six biological replicates) were tested by using RNA-seq. Total RNA was extracted with the RNAprep Pure Plant Kit (Tiangen, China) and detected on a 1% agarose gel.” What tissue was used here? Was it root tissue? This is important, because the qRT-PCR validation of the RNA-seq data is done with root tissue. It is stated later (in the results) that these were root samples, but this type of detail should definitely be included in the methods section.

Lines 119-120 of the Methods. It is stated that “The purity, concentration, and integrity of the
total RNA samples were assessed prior to further analysis.” How was this done? Qubit for concentration? Tape Station or similar to look at integrity? This information should be specified.

Lines 121-125 of the Methods. It is stated that “In this work, twelve cDNA libraries
produced from the ACS and AC were sequenced by using the Illumina HiSeqTM 4000 platform and paired-end reads were generated, and clean reads were obtained by removing raw reads containing adapter, reads containing ploy-N and lower quality reads. Transcriptome assemblies for the twelve libraries were performed separately using Trinity (Garber et al., 2011).”. Again, there is insufficient information here to gauge exactly what was done. What kind of adapter sequences were used? Was a library prep kit used? If so, what kind? Was PCR amplification used during the library prep? How long were the paired-end reads? What settings were used with Trinity?

Lines 129-132 of the Methods. It is stated that “We used KOBAS 2.0 (Xie et al.,
2011 ) software to test the statistical enrichment of DEGs in KEGG pathways. After further investigation, we selected the flavonoid biosynthesis pathways for more detailed analysis.” What was the “further investigation” that was done here. In other words, what were the criteria for determining that the flavonoid pathway would be further investigated?

Line 163 of the Methods. It is stated that “The KEGG database was employ to analyze the metabolites and metabolic pathways.” Again, this is not enough information for a reader to really understand or judge what was done. How was KEGG used to analyze pathways? Some degree of detail is needed here.

Line 165 of the Methods. It is stated that “We performed conjoint analyses on DEGs and DEMs to ascertain the degree pathways.” What does this mean? Again, need more detail. What are “degree pathways”? How are they relevant to the analysis? What tools were used for this analysis?

Lines 165-167 of the Methods. It is stated that “To uncover the regulatory mechanism of flavonoid biosynthesis, we measured the relevance between metabolome and gene by calculating the Correlation Coefficient (CC). ” What correlation coefficient was used here? Spearman? Pearson? How was it calculated? It’s not clear from this statement what sets of values are being correlated. Likewise, the canonical correlation analysis is not explained in adequate detail.

Validity of the findings

Lines 241-254 of the Results (sub-header “Correlation analysis between RNA-seq and metabolite uncovers the regulatory of flavonoid biosynthesis”) is stated to have the goal to “unveil the underlying regulatory mechanism in flavonoid biosynthesis metabolism between ACS and AC, we performed correlation analyses of the metabolites related to flavonoid biosynthesis and the transcripts.” I do not think that the results presented in this section reach this goal. They certainly demonstrate that there are correlations between metabolites and gene expression that make sense based on what we know about the structure of the flavonoid pathway. However, there is no clear demonstration of a regulatory mechanism, rather there are only correlations. Furthermore, these correlations don’t have any clear evidence of directionality or interrelationships between genes/metabolites (i.e. co-regulatory relationships), which we know do exist systems where flavnoid regulation has been looked at in some detail. This is not to say that the correlation analysis here does not have value, but rather that it doesn’t meet the stated goal of unveiling the underlying regulatory mechanism. Doing this would require some sort of functional experiment. So, I think that the language discussing the idea of an "underlying mechanism" should be tempered and reflect that only correlations between gene expression and metabolite profile is shown here.

---

## Round 0.2 · Minor Revisions

Dear authors,

Thank you for your revision. However, our reviewer still had minor comments and suggestions to make the manuscript suitable for publication.

Best regards

·

Basic reporting

The article is greatly improved from the first version. The data and methods are communicated much more clearly and the language has been clarified. I still feel that it would be better to show the names of the flavonoid pathway genes in Figures 8 and 9, rather than the alphanumeric Unigene codes. Not having the gene names makes it much harder to interpret and relate to what is going on and what is being said in the text. I am also still confused by the volumes described in lines 137-145. It seems to me that if these were pipetted into a "microplate", that the units should be uL (microliters) and not mL (millileters).

Experimental design

no comment

Validity of the findings

I think the findings are valid and the justification and interpretation of the results is now much clearer.

Additional comments

I think that this is now publishable with some minor revisions to clarify issues that I mention above.

---

## Round 0.3 · Minor Revisions

Dear Dr. Liu and co-authors,


Thank you for your revision. However, our Section Editor has commented as follows.

"1. Raw reads and assembled unigenes need to be deposited in a public repository.


2. Please clarify: what was input to Deseq: raw reads or FPKM?


3. Please clarify: same sample used for RNAseq and metabolites?


4. Was any filtering or collapsing done after Trinity assembly to obtain unigenes? If so, please describe.


5. “Rich factor”. should be “enrichment factor” and not capitalized.


6. Citation needed for DEseq2"


Would you please revise the manuscript?

Best regards

·

Basic reporting

The modifications made to the manuscript have fixed the points that I found confusing and made the presentation clearer. Adding the gene names to figures 8 and 9 make them much easier to interpret. I think the manuscript is now suitable for publication.

Experimental design

NA

Validity of the findings

NA

Additional comments

NA

---

## Round 0.4 · Minor Revisions

Dear authors,

Thank you for your revision. However, our Section Editor has commented and said:

"Thanks for the responses.

Two issues remain:

1) DEseq requires raw readcounts, not FPKM. See http://bioconductor.org/packages/devel/bioc/vignettes/DESeq2/inst/doc/DESeq2.html#why-un-normalized-counts

2) unigenes need to be deposited to a public repository. NCBI Transcript Shotgun Assembly database will work for this: https://www.ncbi.nlm.nih.gov/genbank/tsa/";

Would you please consider the comments?

Best regards

---

## Round 0.5 · accepted · Accept

Dear authors,

Thank you for your revision.

Best regards